# The Immunotherapy for Colorectal Cancer, Lung Cancer and Pancreatic Cancer

**DOI:** 10.3390/ijms222312836

**Published:** 2021-11-27

**Authors:** Shiu-Jau Chen, Shao-Cheng Wang, Yuan-Chuan Chen

**Affiliations:** 1Department of Neurosurgery, Mackay Memorial Hospital, Taipei 104217, Taiwan; chenshiujau@gmail.com; 2Department of Medicine, Mackay Medical College, New Taipei City 252005, Taiwan; 3Department of Psychiatric, Taoyuan General Hospital, Ministry of Health and Welfare, Taoyuan 33004, Taiwan; 4Department of Mental Health, Johns Hopkins Bloomberg School of Public Health, Baltimore, MD 21205, USA; 5Jenteh Junior College of Medicine, Nursing and Management, Hou-Loung Town, Miaoli County 35664, Taiwan; 6Program in Comparative Biochemistry, Interdisciplinary Graduate Group, University of California, Berkeley, CA 94720, USA

**Keywords:** colorectal cancer, lung cancer, pancreatic cancer, immunotherapy, immune checkpoint, monoclonal antibody

## Abstract

Immunotherapy is a novel anti-cancer method which employs a different mechanism to conventional treatment. It has become a significant strategy because it provides a better or an alternative option for cancer patients. Recently, immunotherapy has been increasingly approved for the treatment of cancer; however, it has various limitations; for instance, it is only suitable for specific patients, the response rate is still low in most cases, etc. Colorectal cancer, lung cancer and pancreatic cancer are known as three major death-causing cancers in most countries. In this review, we discuss immunotherapeutic treatment for these three cancers, and consider the option, prospects and limitations of immunotherapy. The development of immunotherapy should focus on the discovery of biomarkers to screen suitable patients, new targets on tumors, neoadjuvant immunotherapy and the combination of immunotherapy with conventional therapeutic methods. We can expect that immunotherapy potentially will develop as one of the best therapies for patients with advanced cancer or poor responses to traditional methods.

## 1. Introduction

Immunotherapy is an anti-cancer method employing a mechanism that is significantly different from traditional therapeutics. It has become an important strategy for the clinical treatment of cancers. Approval of immunotherapeutic drugs has been increasing, with various treatments in clinical and preclinical development [1]. The principle of these drugs for immunotherapy includes the examination of one’s own immune system, the engineering/reeducation of T cells to recognize cancer cells and further to attack them or the adding of inhibitors to block T cell receptors/tumor cell ligands. Immunotherapy can be classified into active immunotherapy, passive immunotherapy and combined immunotherapy. Active immunotherapy directly induces the autoimmune system so that it can recognize specific antigens on cancer cells and attack tumors. Passive immunotherapy uses exogenous substances to exert anti-tumor effects, including monoclonal antibodies, lymphocytes, cytokines, etc. Combined immunotherapy is the combined use of active/passive immunotherapy and traditional therapeutics.

The immune checkpoint is a group of membrane proteins (receptors) expressing on effector cells (e.g., T cells, B cells, NK cells), consisting of multiple co-inhibitory and co-stimulatory pathways. It participates in the elimination of unwanted substances while ensuring self-tolerance, which plays an important role in immunomodulation. Tumor cells containing specific ligands are often able to bind to specific receptors to activate inhibitory checkpoint pathways and evade immune responses. The immune checkpoint executes a regulatory mechanism which in healthy people makes the immune function of T cells maintain a normal and balanced state by regulating the action of ligands and receptors. When T cells are activated, they will express more immune checkpoint receptors, such as programmed cell death protein 1 (PD-1) or cytotoxic T lymphocyte-associated antigen 4 (CTLA-4) [2,3]. When these receptors bind to inhibitory ligands, the activity of T cells will be inhibited to avoid excessive immune responses that may damage normal cells and healthy tissues.

Cancer cells have many neoantigens due to many kinds of mutations. In theory, these neoantigens should be recognized by the immune system and activate T cells to destroy cancer cells. However, cancer cells continue to survive and proliferate, indicating that cancer cells can escape the surveillance of the immune system. Most cancer cells producing neoantigens can really be eliminated by T cells and only some cancer cells are capable of avoiding the host immune system. Recent studies have shown that cancer cells can use the mechanism of immune checkpoints to attenuate the activity of T cells [2]. For example, lung cancer cells can express more programmed cell death protein ligand 1 (PD-L1) and binds to PD-1 receptors to inhibit the immune function of T cells. However, the antitumor activity of T cells will be initiated if the inhibitors for PD-L1 or PD-1 bind to the PD-L1 ligand or PD-1 receptor, respectively (Figure 1). A similar inhibition reaction is also found in CTLA-4 receptors on T cells, and other potential targets, such as B and T lymphocyte attenuator (BTLAs) [4], the variable domain immunoglobin suppressor of T cell activation (VISTA), the T cell immunoglobulin and mucin-containing protein 3 (TIM3), the lymphocyte-activated gene-3 (LAG-3, CD223) and CD47 [5]. Additionally, there are agonists of costimulatory molecules to enhance the immune checkpoint signaling in the tumor microenvironment, such as 4-1BB (CD137), OX40 (a member of the tumor necrosis factor receptor superfamily 4, CD134), glucocorticoid-induced tumor necrosis factor receptor (GITR, a type I transmembrane protein), inducible T cell costimulator (ICOS), CD40 and CD28 [6]. Based on these mechanisms, immune checkpoint inhibitors show promise to be developed as drugs for immunotherapy, and there have been many immune checkpoint inhibitors approved by the United States Food and Drug Administration (U.S. FDA) for the treatment of cancer (Table 1).

In early 2017, colorectal cancer, lung cancer and pancreatic cancer were ranked the first three leading causes of cancer-related death in the U.S. [7]. The increasing trend in incidence and mortality of these three cancers is expecting to be similar in the U.S. and the world for the near future. Therefore, we focus on reviewing the immunotherapies for colorectal cancer, lung cancer and pancreatic cancer in this article.

## 2. Immunotherapy of Colorectal Cancer

Colorectal cancer is a cancer originating from the colon or rectum. Due to the abnormal growth of large intestine cells, it may invade or metastasize to other parts of the body. The occurrence of colorectal cancer is closely related to the growth of polyps, and tumors evolved from polyps are the main cause. The polyps can be divided into three types: proliferative, adenomatous and inflammatory [8,9,10,11]. The conventional treatment of colorectal cancer is based on surgical resection in the early stage and chemical and/or radiotherapy may be combined according to the condition of the disease. The conventional methods are described as follows: (1) minimally invasive surgery [12,13,14]; (2) chemotherapy [12,13,14]; (3) radiation therapy [12,13,14]; (4) targeted therapy [15]; (5) others: trifluridine/tipiracil (Lonsurf^®^) is an orally active antimetabolite agent composed of trifluridine (a thymidine-based nucleoside analogue) and tipiracil (a potent thymidine phosphorylase inhibitor) [16,17,18]. Celecoxib (Celebrex^®^), a cyclooxygenase-2 (COX-2) inhibitor and nonsteroidal anti-inflammatory drug (NSAID), has been used to reduce colorectal polyps in patients with familial adenomatous polyposis. Celecoxib can reduce the risk of recurrent colorectal adenomas, but its effects in reducing the risk of colorectal cancer are still poorly understood [19].

### 2.1. Immunotherapy Using Checkpolint Inhibitors

Mismatch repair (MMR) proteins are responsible for repairing errors that occur in the DNA synthesis process. If these proteins are absent, instability of microsatellites (a short and repetitive DNA sequence) will result. Microsatellite instability (MSI) errors, including unstable and shortened microsatellites, can be detected using immunohistochemistry staining or polymerase chain reaction. The occurrence of shortened and/or unstable microsatellite in microsatellite instability-high (MSI-H) tumors are due to lack of MMR proteins. Clinically, it is found that patients with colorectal cancer must have MSI-H or defective mismatch repair (dMMR) genotypes for immunotherapy to be effective. PD-1 plus CTLA-4 blockade is highly effective in advanced-stage, dMMR colorectal cancers, but not in MMR-proficient (pMMR) tumors [20,21,22]. 

The use of immune checkpoint inhibitors has changed the therapeutic methods and outcomes of many diseases, such as colorectal cancer, lung cancer, kidney cancer, bladder cancer, melanoma, Hodgkin’s lymphoma, etc., but metastatic colorectal cancer (mCRC) rarely responds to this immunotherapy. Nevertheless, the US FDA approved the PD-1 inhibitors, including pembrolizumab and nivolumab, as the first-line treatment for patients with unresectable or metastatic MSI-H or dMMR colorectal cancer in 2020 [23] and 2019 [24], respectively (Table 1), for the treatment of tumors containing MSI-H or in cases of mCRC. Most mCRC patients are not MSI-H and will not benefit from the treatment of immune checkpoint inhibitors. For these patients, the development of predictive biomarkers for personalized medicines and clinical trials are required. Therefore, it is necessary to screen the patients who could potentially benefit from the immune checkpoint blockade.

Llosa et al. studied 26 patients with advanced mCRC treated with pembrolizumab to address whether immunohistology stratification of mCRC was based on primary tumor PD-L1 expression, which was related to the presence or absence of extracellular mucin (a tumor-associated antigen), to define a subset of mCRC patients who show a pre-existing antitumor immune response [25]. They used the incorporation of histopathological characteristics (% extracellular mucin) and PD-L1 expression at the invasive front to derive a CPM score (composite PD-L1 and mucin) to discriminate between patients who showed clinical benefit (complete, partial response or stable disease) and those patients with progressive disease [25]. If the results can be validated in larger cohorts, the CPM score in combination with MSI testing may guide immunotherapy interventions for the treatment of mCRC patients. This method potentially opens up immunotherapy to more mCRC patients by including microsatellite stable mCRC patients who are not captured by current molecular biomarkers [25].

### 2.2. Neoadjuvant Immunotherapy 

Adjuvants are compounds (e.g., inorganic molecules, organic molecules, polymers, and colloids) used alone or in combination with other agents to activate immune responses. They can activate antigen-presenting cells and promote the epitopes to be presented on the major histocompatibility complex class I (MHC-I), further enhancing the activity of cytotoxic T lymphocytes to destroy cancer cells. Ipilimumab is a monoclonal antibody that can activate immune responses by targeting cCTLA-4, a protein receptor that downregulates the immune system. It turns off the inhibition mechanism which cytotoxic T cells exhibit to destroy cancer cells and boosts the body’s immune response against cancer cells. Ipilimumab has been approved by the US FDA in 2011 for the treatment of melanoma [26]. Ipilimumab was also used as a neoadjuvant for colorectal cancer immunotherapy.

Chalabi et al. proposed neoadjuvant immunotherapy for patients with colorectal cancer at early stage [27]. In this study, patients with dMMR or proficient MMR (pMMR) tumors received a single dose of ipilimumab and two doses of nivolumab before surgery; the pMMR group received celecoxib or no treatment. They found the treatment was well tolerated and all patients underwent radical resections on schedule, meeting the primary endpoint. Of the patients who received ipilimumab and nivolumab (20 dMMR and 15 pMMR tumors), all 35 patients were evaluated for efficacy and translational endpoints. In dMMR tumors, all 20 patients showed pathological responses, with 19 major pathological responses and 12 pathological complete responses [27]. In pMMR tumors, four patients showed pathological responses, with three major pathological responses and one partial pathological response [27]. The results revealed that neoadjuvant immunotherapy potentially become the standard of care for a defined group of patients with colorectal cancer when validated in larger studies if at least three years of disease-free survival data are available [27].

### 2.3. Development of New Targets for Immunotherapy

Chemokine (CC motif) receptor 8 (CCR8) is a receptor expressed on regulatory T cells (Treg), which is known to be critical to CCR8 + Treg-mediated immune suppression important. Recent studies have shown that, compared with tissue-resident Tregs in normal tissues, CCR8 is particularly high in tumor-resident Tregs in breast cancer, colorectal cancer and lung cancer patients [28]. Therefore, CCR8 may become a reasonable target for cancer immunotherapy, which is used to regulate tumor-resident Treg, enhance the anti-tumor immune response and prolong the survival of patients.

Villarreal et al. demonstrated that monoclonal antibodies against CCR8 can significantly inhibit tumor growth and improve the long-term survival rate of mice suffering from colorectal tumors in animal study [28]. This anti-tumor activity is closely associated with the increase of tumor-specific T cells, the enhancement of CD4 + T cells and CD8 + T cell infiltration and the significant decrease of tumor resident CD4 + CCR8 + Treg. The results showed that CCR8 has the potential to be a new target for colorectal cancer immunotherapy. Targeting of CCR8 can effectively reduce the resident Treg of colorectal tumors [28]. This finding indicated that CCR8 blockade or depletion antibody may be used for the clinical treatment of a variety of cancers, including colorectal cancer, whether it is used as a monotherapy or in combination with other immunotherapies [28].

## 3. Immunotherapy of Lung Cancer

Lung cancer can be divided into non-small cell lung carcinoma (NSCLC, 85%) and small cell lung carcinoma (SCLC, 15%). The former mainly includes adenocarcinomas, squamous cell carcinomas and large cell carcinomas, of which lung adenocarcinoma is the most common, accounting for more than 90% of cases [29,30,31]. There are almost no symptoms in the early stage of lung cancer. The most common way to detect lung cancer in clinical practice is X-ray examination. However, it is difficult to detect tumors smaller than 1 cm because of low sensitivity. Studies have shown that the cure rate of lung cancer with surgical resection of about 1 cm is about 85% to 90% (without recurrence in five years). Low-dose computed tomography is currently the most sensitive method for lung cancer detection. Its detection of lung lesions as small as 0.3 cm is helpful for early diagnosis and follow-up treatment [31,32,33]. Depending on different types, stages, symptoms and severity, various conventional therapeutic strategies can be used, including surgical treatment, chemotherapy, radiotherapy and targeted therapy [34,35,36,37,38,39,40,41].

### 3.1. Immunotherapy Using Checkpolint Inhibitors

The immunotherapy of lung cancer uses monoclonal antibodies as checkpoint inhibitors to bind to tumor cell ligands or immune cell receptors. It reduces the inhibitory effect of tumor cells on immune cells and restores the ability of the immune system to attack lung cancer cells to achieve the therapeutic effect [41,42,43,44,45]. The advantages include a lower occurrence of serious side effects compared to traditional treatments, a therapeutic effect longer than those of traditional treatments for specific patients and the feasibility of combination with traditional treatments. The disadvantages include the need to detect specific biomarkers to identify whether the patient is suitable and the possibility of triggering an autoimmune response, and more research is required to confirm its efficacy, as immunotherapy is still in the early stage of development [42,43,44,45].

Karak et al. evaluated the efficacy of monoclonal antibodies (nivolumab and pembrolizumab) that inhibit PD-1/PD-L1 when used above the second line [46]. They included 110 patients who had undergone first-line standard chemotherapy and had developed into stage 4 NSCLC and had received PD-1 inhibitory drugs in the second line or above. It was found that among 44 patients who underwent immunohistochemical tests, the PD-L1 performance rate was higher than 50%; 1–49% and less than 1% were 17 (38.6%), 12 (27.3%) and 15 respectively (34.1%). Of patients, 74.7%, 21.8%, and 3.5% using monoclonal antibodies that inhibit PD-1/PD-L1 as second, third and fourth-line treatment drugs, respectively. Observing the tumor response to the treatment, 43.4% of patients were found to continue to deteriorate (progressive disease), 31.3% had stable disease, 22.2% had a partial response and 3.1% had a complete response [46]. They observed the survival period of patients to have a median of progression-free survival (PFS) to be 4 months, and the median of overall survival (OS) is 8.1 months while using these drugs for second-line treatment. The median PFS and OS were 3.1 and 7.8 months, respectively, when these drugs were used in the third line. The study showed that immunotherapy has the potential to become a tool for lung cancer oncology and has significant responses in clinical treatment [46].

### 3.2. Neoadjuvant Immunotherapy

Jia et al. used major pathological response (MPR) and pathological complete response (PCR) to evaluate the efficacy of neoadjuvant immunotherapy for 252 patients with resectable NSCLC [44]. They found that the values of MPR using neoadjuvant immunotherapy were significantly higher (MPR: odds ratio (OR) = 0.59; confidence interval (Cl) 95%, 0.36–0.98; pCR: OR = 0.16; 95% CI, 0.09–0.27), compared with neoadjuvant chemotherapy (less than 25% MPR and 2–15% PCR) [47]. Additionally, they evaluated the safety of immunotherapy by the incidence of treatment-related adverse events, surgical resection rate, incidence of surgical complications and surgical delay rate, and the pooled OR values were demonstrated to be better than those for neoadjuvant chemotherapy. The mean surgical resection rate was 88.70%, which was similar to neoadjuvant chemotherapy (75–90%). The results show that neoadjuvant immunotherapy is more effective and safer than that of chemotherapy for resectable NSCLC, though no dominant immune checkpoint inhibitors were used in neoadjuvant immunotherapy [47].

## 4. Immunotherapy of Pancreatic Cancer

The pancreas has the function of exocrine and endocrine glands. It secretes pancreatic juice (containing enzymes) and hormones which are needed for digestion and maintaining carbohydrate and growth balance in the body. The preliminary statistics show that pancreatic cancer caused by cancer cells in the pancreatic islets accounted for about 5–10% of cases, while pancreatic duct adenocarcinoma accounted for about 90–95% [48,49,50]. Compared with other cancers (e.g., lung cancer, colorectal cancer, breast cancer, cervical cancer, etc.), pancreatic cancer has lower incidence but up to 85% of patients are clinically diagnosed at advanced stages and cannot undergo surgery [51,52]. Even if there is a chance for surgery, many patients will still relapse, and the diagnosis is almost equivalent to a declaration of death. Consequently, pancreatic cancer is referred as the “King of Cancer” because of the difficulty of detecting it early on and because the efficacy of treatment is poor and the recurrence rate high [51,52,53]. At present, the most effective method for pancreatic cancer treatment is surgical removal, but only about 15% of patients are clinically suitable for surgery. For patients who are unable to undergo surgery, only chemotherapy, radiotherapy, targeted therapy or combined therapy can be used to relieve the disease and prolong the survival period, but the efficacy is usually not ideal [53,54,55,56].

Immunotherapy is another option for pancreatic cancer patients who respond poorly to conventional methods. There have been many clinical trials trying to evaluate the effectiveness of immunotherapy for pancreatic cancer; however, current studies have failed to change the clinical outcome significantly. In addition, the identification of pancreatic tumor-associated antigens, which functionally contribute to pancreas pathogenesis, and their successful implication in cancer treatment is still challenging. Fortunately, mucin 4 (MUC4), a glycoprotein with a high molecular weight, seems to be a novel and attractive tumor-associated antigen, because it is overexpressed in mouse and human pancreatic tumors, not being detected in the normal pancreas [57]. The recombinant MUC4 domain and predicted immunogenic T cell epitopes can induce cell-mediated and humoral immune responses against MUC4, suggesting its potential to be used as a vaccine candidate for the treatment of pancreatic cancer. Additionally, immunotherapy potentially has the synergistic effect of increasing the response rate and combining with other conventional therapies [56,58]. Several agents for single use for immunotherapy are under study as follows.

### 4.1. Immune Check Point Inhibitor

Immune checkpoint blockers (e.g., anti-CTLA-4, anti-PD-1, anti-PD-L1) have shown curative effects in some malignant tumors, but no clinical trials have proved any efficacy in most cases of pancreatic cancer at stages I and II. However, a combination therapy combining immune checkpoint inhibitors and radiotherapy and/or chemotherapy has initially shown positive results [59,60,61].

Group 2 innate lymphoid cells (ILC2s) found in cancers of mammal tissues is known to be able to modulate inflammation and immunity, but the role of ILC2 in cancer immunity and immunotherapy is still poorly understood. Moral et al. demonstrated that tissue-specific tumor immunity was activated by ILC2s infiltrated from pancreatic ductal adenocarcinomas [62]. In animal study, they found that interleukin-33 (IL33) activates tumor ILC2s and CD8^+^ T cells to limit pancreas-specific tumor growth in orthotopic pancreatic tumors but not heterotopic skin tumors [62]. The tumor ILC2s express the inhibitory checkpoint receptor PD-1. The PD-1 blockade decreases ILC2 cell-intrinsic PD-1 inhibition to expand tumor ILC2s and increase immune responses to control tumor growth, indicating that activated tumor ILC2s may be targets of anti-PD-1 immunotherapy. The results showed that ILC2s are anti-cancer immune cells for pancreatic cancer immunotherapy because they can be used as tissue-specific enhancers that amplify anti-PD-1 efficacy [62]. The immunotherapy strategy to collectively target anti-cancer ILC2s and T cells is potentially applicable in the treatment of pancreatic cancer.

### 4.2. Therapeutic Cancer Vaccine

Therapeutic cancer vaccines (e.g., whole cells, dendritic cells, DNA and peptide vaccines) can stimulate the presentation of immunogenic cancer antigens to the immune system. A vaccine using a single peptide derived from the tumor-associated self-antigen human telomerase demonstrated no response or survival benefit in patients with metastatic pancreatic cancer in a phase III trial [63]. However, these vaccines have the potential to activate cancer antigen-specific cytotoxic T lymphocytes and trigger subsequent anti-cancer immune responses [59,60,63].

Tumor-derived exosomes (TEXs) are lipid nanoparticle encapsulated vesicles that transport bioactive substances into the microenvironment to promote tumor progression. Nonetheless, many recent studies show that TEXs can efficiently enhance immune responses against tumors if they are given appropriately [64]. Naseri et al. summarized in a review the potency of TEXs in inducing effective anti-tumor responses in vitro and preclinical studies [64]. The recently modified strategies further improve TEX vaccination efficacy. Although it is required for TEXs to have further experimental studies to determine efficacy and side effects, they indicated that TEXs are promising to become new objects in cancer vaccination based on tumor antigen-selective high immunogenicity [64]. There may be a little hope that TEXs will prove a breakthrough in tumor immunotherapy.

### 4.3. Adoptive Cell Transfer

The patient’s own tumor antigen-specific T cells were collected and genetically modified. These cells were cultivated and proliferated in vitro, and then re-transplanted into the patient’s body to enhance immunity and improve immune responses. Engineered chimeric antigen receptor T (CAR-T) cell therapy is clinically the most common type of adoptive cell transfer therapy [59,60]. Unfortunately, translating CART therapy to malignancies is still challenging, because it is difficult to identify a safe, specific and homogeneously expressed target [65]. However, several self-antigens, such as carcinoembryonic antigen, prostate stem cell antigen, mesothelin and human epidermal growth factor receptor 2, are significantly overexpressed in pancreatic duct adenocarcinomas and are related to worse prognoses [66], and therefore likely to be prospective targets.

### 4.4. Agonistic Immunotherapy

Antigen presentation cell (APC) activators and T cell activators are two methods under investigation. It is known that CD40 targets modulate the activation of APC and finally result in T cell activation (Figure 2). Agonists of costimulatory molecules like CD40 have shown promising results in preclinical studies and are currently being tested in ongoing clinical trials [67]. Activating CD40 therapy can mimic the assistance of T cells and allow APCs to be more competent to effectively present antigens to T cells and activate them. No enhancement of long-term survival was observed in phase I clinical trials, suggesting the therapy did not induce immune memory [68]. However, the combination of activating CD40 monotherapy and gemcitabine can activate macrophages to kill cancer cells and has shown efficacy in the clinical trial [59,60].

### 4.5. Myeloid-Based Immunotherapy

Several subtypes of myeloid cells derived from bone marrow (e.g., macrophages, dendritic cells, neutrophils, monocytes and granulocytes, etc.) significantly regulate the growth and progression of tumors via the supplement of tumor-promoting factors and molecules that suppress CD8^+^ cytotoxic T cells [69]. Macrophages and monocytes are considered the most populous myeloid lineage cells in developing solid tumors (Figure 3) and play a crucial role in regulating both protumor and antitumor immune responses. Targeting of these cells potentially attenuates solid tumor progression by the induction and mobilization of cytotoxic T cells [69]. The abnormal immune response of pancreatic cancer is partly regulated by immunosuppressive bone marrow; suppressing the bone marrow can suppress the tumor. The bone marrow is controlled by cytokines, chemokines and signaling molecules. The receptors of cytokines, chemokines and signaling molecules can establish the immunosuppressive tumor microenvironment and potentially be used as therapeutic targets [59,60].

Zhang et al. determined the effect of myeloid cell depletion on the onset and progression of pancreatic cancer in mice with CD11b-diphtheria toxin receptors [70]. They found that myeloid cells inhibit cytotoxic T cell activity by inducing the expression of PD-L1 in tumors depending on the production of epidermal growth factor receptors (EGFR)/mitogen-activated protein kinases (MAPK). The results demonstrated that myeloid cells support immune evasion in pancreatic cancer in an EGFR/MAPK-dependent regulation of PD-L1 expression on tumors [70]. This study revealed the relationship between myeloid cells and T cell-mediated immunity within the pancreatic cancer microenvironment. The suppression of myeloid cells is able to restore the anti-tumor activity of cytotoxic T cells— a new finding with implications for the establishment of immunotherapy for pancreatic cancer [70].

### 4.6. Stroma-Modulating Immunotherapy

The tumor microenvironment is the environment surrounding the tumor, including blood vessels, immune cells, fibroblasts, signaling molecules and the extracellular matrix [71,72]. Tumors and the microenvironment are so closely related that tumors can influence the microenvironment via the release of extracellular signals, the promotion of tumorigenesis and the triggering of neighboring immune tolerance (Figure 4). The immune cells in the microenvironment can affect the growth and evolution of cancer cells and tumorigenesis is regulated by the microenvironment [71,72]. The proliferative matrix of pancreatic cancer is a key component of the immunosuppressive tumor microenvironment and an obstacle to effective treatment. Although it is still controversial whether targeting of the matrix is beneficial for patients, early-stage studies proving the therapeutic potential of modulating substrates have already begun [54,55,73].

## 5. Discussions

It is known that not all patients with cancer are suitable for immunotherapy; for example, colorectal cancer patients who can be treated with immunotherapy must have MSI-H or dMMR under the condition of tumor cell genotype classification. For immunotherapy, there are not many suitable patients, and the current clinical response rate is not satisfactory. However, with a better understanding of tumor molecular biology and the development of new targets, immunotherapy is expected to open up new possibilities for cancer therapy and allow for disease recovery, providing fresh treatment options for patients who have responded poorly to traditional treatments.

The development of immunotherapy has provided new or alternative options for the treatment of patients with cancer, bringing the treatment of cancer closer to precision medicine. Immune checkpoint inhibitors are current immunotherapies that have been proven effective for the treatment of cancer. These therapies sometimes perform better than traditional chemotherapy, with curative effects and fewer side effects. However, the immune checkpoint inhibitors currently approved for the treatment of cancer have very low response rates (e.g., only about 20% for NSCLC) in patients who have not been screened by biomarkers. Since adjuvant therapy and neoadjuvant therapy are always complementary or combined, neoadjuvant immunotherapy using a standard treatment strategy is still challenging. Therefore, the improvement of the patient’s response to immunotherapy should focus on the development of biomarkers for screening suitable patients, the discovery of new targets on tumors, the finding of neoadjuvants or the combination of immunotherapy with one (or more) different conventional methods, and there are currently many clinical trials in progress. Like targeted therapy, immunotherapy is not suitable for everyone due to different tumor characteristics, but the most suitable treatment can be found after testing. It can be expected that immunotherapy will become one of the best therapies for patients with advanced cancer, including colorectal cancer, lung cancer and pancreatic cancer, or poor responses to traditional treatments.

## Figures and Tables

**Figure 1 ijms-22-12836-f001:**
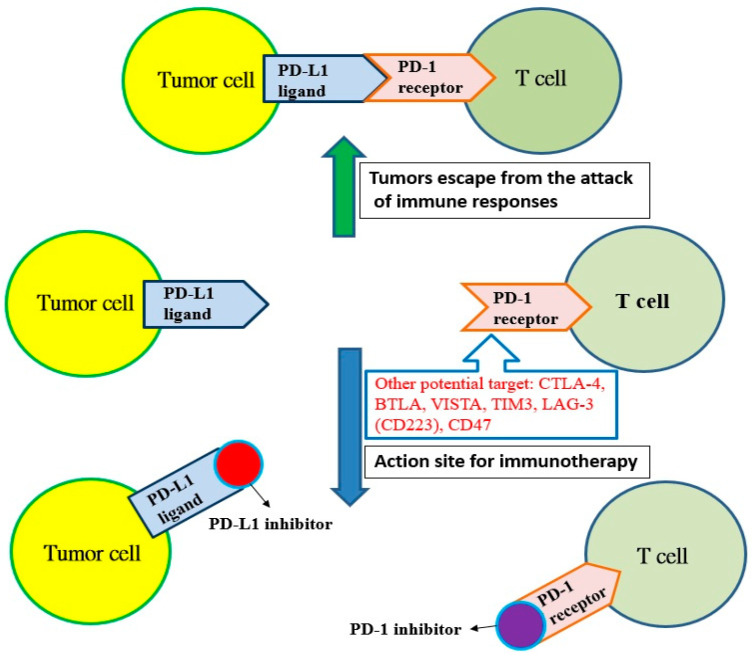
The programmed cell death protein ligand 1 (PD-L1) of tumor cells binds with the programmed cell death protein 1 (PD-1) receptor on T cells, and tumors escape from the attack of immune responses. However, T cells can recognize tumor cells and initiate immunotherapy if the PD-1 receptor is blocked by the PD-1 inhibitor or the PD-1 ligand is blocked with the PD-L1 inhibitor. Other potential targets: cytotoxic T lymphocyte-associated antigen 4 (CTLA-4); B and T lymphocyte attenuator (BTLA); variable domain immunoglobin suppressor of T cell activation (VISTA); T cell immunoglobulin and mucin-containing protein 3 (TIM3); lymphocyte-activated gene-3 (LAG-3, CD223); CD47.

**Figure 2 ijms-22-12836-f002:**
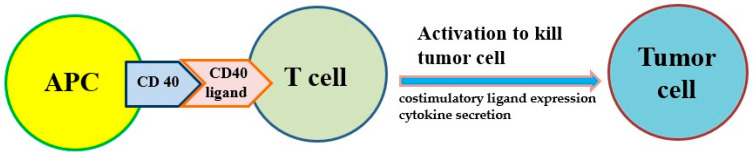
The costimulatory receptor CD40 on antigen-presenting cells (APCs) can improve the antitumor response of T cells because it induces costimulatory ligand expression and cytokine secretion that drive antitumor activity.

**Figure 3 ijms-22-12836-f003:**
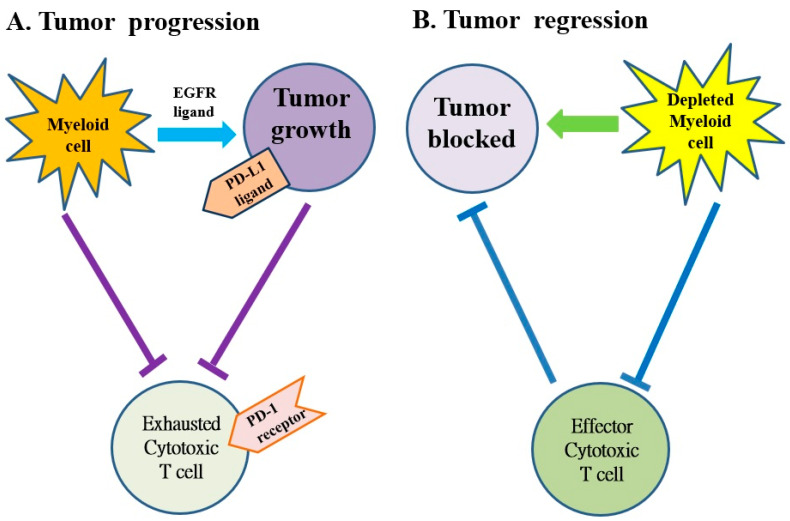
The myeloid cells protect tumor cell viability by blocking the anti-tumor responses of cytotoxic T cells in pancreatic cancer. (**A**) The myeloid cells block anti-tumor immune responses of cytotoxic T cells by activating the programmed cell death-1 (PD-1)/PD-ligand 1 (PD-L1) checkpoint. (**B**) The myeloid cell depletion reverses immune suppression and activates CD8+ T cells to block the growth of tumors. EGFR: epidermal growth factor receptor.

**Figure 4 ijms-22-12836-f004:**
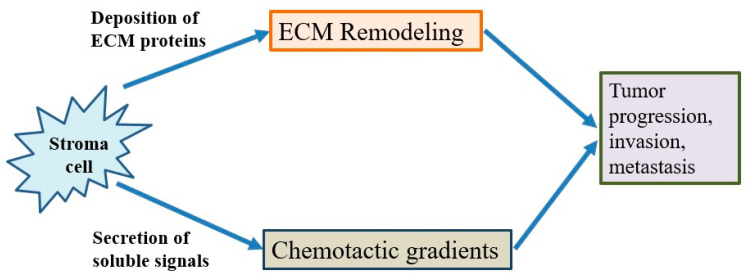
The behavior of cancer cells is affected by their environment. The stromal cells are able to release chemotactic growth factors, and cell-induced mechanical strains are able to rearrange extracellular matrix (ECM) fibers. These factors are correlated with tumor progression, invasion and metastasis. In addition, the tumor cell interacts with fibroblasts to lead to the deposition of new ECM proteins, and physical forces from strains are related with fiber alignment, resulting in persistent migration and invasion of cancer cells.

**Table 1 ijms-22-12836-t001:** The monoclonal antibodies approved by the U.S. FDA to be used as immune checkpoint inhibitor for immunotherapy related to lung cancer or colorectal cancer.

Immune Checkpoint Inhibitors	Mechanism	Indication
Pembrolizumab (Keytruda^®^)	Inhibition of programmed cell death protein (PD-1)	Lung cancer, head and neck cancer, Hodgkin lymphoma, stomach cancer, colorectal cancer,
Nivolumab (Opdivo^®^)	Inhibition of PD-1	melanoma, lung cancer, malignant pleural mesothelioma, renal cell carcinoma, Hodgkin lymphoma, head and neck cancer, urothelial carcinoma, colonrectal cancer, esophageal squamous cell carcinoma, liver cancer, gastric cancer and esophageal or gastroesophageal junction cancer.
Atezolizumab (Tecentriq^®^)	Inhibition of programmed cell death protein ligand 1 (PD-L1)	Urothelial carcinoma, non-small cell lung cancer (NSCLC), triple-negative breast cancer (TNBC), small cell lung cancer (SCLC) and hepatocellular carcinoma (HCC).
Durvalumab (Imfinzi^®^)	Inhibition of PD-L1	Certain types of bladder cancr, lung cancer.

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
