# Peer review of "The Immunotherapy for Colorectal Cancer, Lung Cancer and Pancreatic Cancer"

_ijms, 2021, doi:10.3390/ijms222312836_

Round 1
Reviewer 1 Report
The authors provided an extensive list of cancer immunotherapies. I find that this review will be informative to readers entering into this field. To improve the clarity, I recommend to add more diagrams
- line 319, agonistic immunotherapy. Please make an illustration of how activating CD40 therapy works
- line 327, myeloid-based immunotherapy. Please make an illustration of how myeloid cells (macrophages, monocytes, neutrophils) regulate the growth of tumor cells, and how to target them.
- line 351, stroma-modulating immunotherapy. Please make an illustration of how to target stromal components of the tumor.
Author Response
The authors provided an extensive list of cancer immunotherapies. I find that this review will be informative to readers entering into this field. To improve the clarity, I recommend to add more diagrams
- line 319, agonistic immunotherapy. Please make an illustration of how activating CD40 therapy works
Ans: We have added a figure (Figure 2) to illustrate how activating CD40 therapy works.
- line 327, myeloid-based immunotherapy. Please make an illustration of how myeloid cells (macrophages, monocytes, neutrophils) regulate the growth of tumor cells, and how to target them.
Ans: We have added a figure (Figure 3) to illustrate how myeloid cells regulate the growth of tumor cells, and how to target them.
- line 351, stroma-modulating immunotherapy. Please make an illustration of how to target stromal components of the tumor.
Ans: We have added a figure (Figure 4) to illustrate how to target stromal components of the tumor.
Reviewer 2 Report
This manuscript, written by Dr. Chen, review type, with the title of “The Immunotherapy for Colorectal Cancer, Lung Cancer and Pancrea Cancer” focuses on recent data regarding three types of neoplasia and the use of immune checkpoint inhibitors of PD-1, PD-L1, and CTLA4 (among other). This manuscript focuses of the 3 types of cancer that will be more relevant in the future, as pancreatic cancer is rare, but very aggressive.
This review manuscript is well written, it is easy to read, and the information has logic. There is only one figure, for PD-L1 and PD-1.
To improve the manuscript, I recommend the following points:
1- Could you please revise all the text for typhos (for example pancrea”.
2- Could you please revise the abbreviations? I would recommend not to overuse them as it makes it harder for the reader.
3- This review focuses mainly in PD1/L1 and CTLA4, but there are other targets. In my opinion, the authors could expand the section that describes the principles of cancer immunotherapy, and a more complex figure could be added.
For example, checkpoint inhibitor immunotherapy includes PD-1 and PD-L1/2 and CTLA-4 but other potential targets are BTLA, VISTA, TIM3, LAG3, and CD47. Additionally, there are agonists of costimulatory molecules, including 4-1BB (CD137), OX40 (CD134), GIRT, ICOS, CD40, and CD28.
https://www.cellsignal.jp/pathways/immune-checkpoint-signaling-pathway
Other therapeutic approaches include manipulating T cells (chimeric antigens, ex vivo expansion of TILs, CD3-directed therapies), oncolytic viruses, therapies directed to other types of cells of the microenvironment (NK, TAMs, IDO), vaccines, etc.
Of note, if the focus is on lung, colorectal and pancreas, not all these possibilities are currently available.
4- Other types of neoplasia could be mentioned in an additional section, or a table could be added if the authors think it is necessary.
Author Response
This manuscript, written by Dr. Chen, review type, with the title of “The Immunotherapy for Colorectal Cancer, Lung Cancer and Pancrea Cancer” focuses on recent data regarding three types of neoplasia and the use of immune checkpoint inhibitors of PD-1, PD-L1, and CTLA4 (among other). This manuscript focuses of the 3 types of cancer that will be more relevant in the future, as pancreatic cancer is rare, but very aggressive.
This review manuscript is well written, it is easy to read, and the information has logic. There is only one figure, for PD-L1 and PD-1. To improve the manuscript, I recommend the following points:
- Could you please revise all the text for typhos (for example pancrea).
Ans: We have already revised pancrea cancer into pancreatic cancer all over the manuscript.
- Could you please revise the abbreviations? I would recommend not to overuse them as it makes it harder for the reader.
Ans: We have already revised the abbreiation in the manuscript to avoid overusing them.
- This review focuses mainly in PD1/L1 and CTLA4, but there are other targets. In my opinion, the authors could expand the section that describes the principles of cancer immunotherapy, and a more complex figure could be added.
Ans: We have revised Figure 1 into a more complex figure.
For example, checkpoint inhibitor immunotherapy includes PD-1 and PD-L1/2 and CTLA-4 but other potential targets are BTLA, VISTA, TIM3, LAG3, and CD47. Additionally, there are agonists of costimulatory molecules, including 4-1BB (CD137), OX40 (CD134), GIRT, ICOS, CD40, and CD28.
https://www.cellsignal.jp/pathways/immune-checkpoint-signaling-pathway
Other therapeutic approaches include manipulating T cells (chimeric antigens, ex vivo expansion of TILs, CD3-directed therapies), oncolytic viruses, therapies directed to other types of cells of the microenvironment (NK, TAMs, IDO), vaccines, etc.
Of note, if the focus is on lung, colorectal and pancreas, not all these possibilities are currently available.
Ans: We have added some sentences to expand the section that describes the principles of cancer immunotherapy. We only want to focus on immunotherapy of lung cancer, colorectal cancer and pancreatic cancer; thus, we do not discuss all possibilities that are not currently available.
- Other types of neoplasia could be mentioned in an additional section, or a table could be added if the authors think it is necessary.
Ans: We only want to focus on reviewing immunotherapy on lung cancer, colorectal cancer and pancreatic cancer in this manuscript; therefore, it is not necessary to add an additional section to mention other types of neoplasia.
Reviewer 3 Report
The article is well written and review an interesting topic. in my opinion the authors should include additional tables/boxes and figures to improve the readability of their article. I do not have additional comments, since clarity and novelty are appropriate.
Author Response
The article is well written and review an interesting topic. in my opinion the authors should include additional tables/boxes and figures to improve the readability of their article. I do not have additional comments, since clarity and novelty are appropriate.
Ans: Thanks for the reviewer’s comment. We have revised the Figure 1 and added more figures (Figure 2, 3, 4).
Round 2
Reviewer 1 Report
The authors adequately addressed the comments.